# Self-Amplifying RNA Viruses as RNA Vaccines

**DOI:** 10.3390/ijms21145130

**Published:** 2020-07-20

**Authors:** Kenneth Lundstrom

**Affiliations:** PanTherapeutics, CH1095 Lutry, Switzerland; lundstromkenneth@gmail.com

**Keywords:** RNA vectors, RNA replicons, RNA vaccines, immune responses, protection against pathogens, protection against tumor challenges

## Abstract

Single-stranded RNA viruses such as alphaviruses, flaviviruses, measles viruses and rhabdoviruses are characterized by their capacity of highly efficient self-amplification of RNA in host cells, which make them attractive vehicles for vaccine development. Particularly, alphaviruses and flaviviruses can be administered as recombinant particles, layered DNA/RNA plasmid vectors carrying the RNA replicon and even RNA replicon molecules. Self-amplifying RNA viral vectors have been used for high level expression of viral and tumor antigens, which in immunization studies have elicited strong cellular and humoral immune responses in animal models. Vaccination has provided protection against challenges with lethal doses of viral pathogens and tumor cells. Moreover, clinical trials have demonstrated safe application of RNA viral vectors and even promising results in rhabdovirus-based phase III trials on an Ebola virus vaccine. Preclinical and clinical applications of self-amplifying RNA viral vectors have proven efficient for vaccine development and due to the presence of RNA replicons, amplification of RNA in host cells will generate superior immune responses with significantly reduced amounts of RNA delivered. The need for novel and efficient vaccines has become even more evident due to the global COVID-19 pandemic, which has further highlighted the urgency in challenging emerging diseases.

## 1. Introduction

RNA-based vaccines have become potentially promising as alternative approaches to conventional vaccine development [1]. In this review, emphasis has been placed on tumor vaccines, but description of vaccines against pathogenic viruses has also been included as more progress has been made in this area, which can serve as encouragement for treatment and prevention of various types of cancer. Particularly, the utilization of mRNA due to its high potency, rapid production and development and safe administration has received plenty of attention. However, progress has been hampered by issues related to instability and low efficiency of in vivo delivery. Recent improvements in the field addressing optimization of mRNA translation and stability, modulation of immunogenicity and progress in mRNA vaccine delivery have contributed to vaccine platforms against infectious diseases and different types of cancers, showing encouraging results in both animal models and clinical trials. In this context, an mRNA vaccine targeting the rabies glycoprotein provided protection against lethal challenges with rabies virus in immunized mice and pigs [2]. In another study, mice and primates immunized with 30 μg and 50 μg of liposome-encapsulated nanoparticles containing mRNA for Zika virus (ZIKV) membrane and envelope glycoproteins, respectively, were protected against ZIKV challenges [3]. Moreover, lipid nanoparticle-encapsulated mRNA expressing influenza A virus hemagglutinin (HA) protected mice from lethal influenza virus challenges and reduced viral titers in ferret lungs [4]. Initial phase I clinical trial results demonstrated robust prophylactic immunity in humans with mild or moderate adverse events only. In another phase I clinical trial, mRNA encoding rabies glycoprotein was administered to healthy volunteers showing safety and a reasonable tolerability profile [5]. Functional antibodies against the viral antigen could be induced and boosted when administered with a needle-free device but not when a needle-syringe was used. In the context of mRNA-based vaccines, self-replicating RNA viruses present an interesting and attractive alternative as described below [6].

The common feature of self-amplifying RNA viruses is their single-stranded RNA (ssRNA) genome embedded in a capsid and envelope protein structure [6]. In the case of alphaviruses and flaviviruses, the ssRNA genome is of positive polarity in contrast to measles viruses (MVs) and rhabdoviruses, which carry a negative sense ssRNA genome. In any case, the RNA genome of self-amplifying RNA viruses can act directly in the cytoplasm without any need of nucleic acid delivery to the nucleus. In the case of positive polarity, the translation can be initiated directly from the incoming ssRNA genome [7], whereas negative sense RNA molecules require the generation of a positive strand RNA template [8]. All self-amplifying RNA viruses initially express their nonstructural genes resulting in the formation of the RNA replication complex (RNA replicon), responsible for extreme RNA replication in infected host cells [7]. It has been estimated that 200,000 copies of RNA are made from a single RNA molecule, providing together with strong subgenomic promoters the basis for extremely high expression levels of viral proteins. This feature has been taken advantage of in expression vectors engineered from self-amplifying RNA viruses, which have been applied for mammalian and non-mammalian cell lines, primary cells and in vivo [9]. Moreover, self-amplifying RNA virus vectors have been used for vaccine development, targeting both infectious diseases and different types of cancers. In this review, the basic function of self-replicating RNA virus vectors is described, and various preclinical and clinical applications are presented.

## 2. Self-Amplifying RNA Virus-Based Expression

Although efficient expression systems have been developed for alphaviruses, flaviviruses, MVs and rhabdoviruses, the differences in the polarity of the ssRNA genome have required alternative engineering. In the case of alphaviruses and flaviviruses, the procedure is straightforward, whereas MVs and rhabdoviruses require the utilization of packaging cell lines and reverse genetics. For simplicity, the focus is here on the illustration of the Semliki Forest virus (SFV)-based expression system, one of the most commonly applied alphavirus expression systems (Figure 1).

Alphaviruses belong to the family of Togaviruses [7]. Particularly, SFV [10], Sindbis virus (SIN) [11] and Venezuelan equine encephalitis virus (VEE) [12] have been used for the engineering of expression vector systems. In this context, engineering of an expression vector carrying the SFV nonstructural protein genes (nsP1-4), where the gene of interest can be inserted downstream of the strong 26S subgenomic promoter allows in vitro transcribed RNA to be directly translated in cell lines and in vivo (Figure 1). Alternatively, replacement of the SP6 or T7 RNA polymerase promoter upstream of the nsP1-4 region with a cytomegalovirus (CMV) promoter permits direct use of plasmid DNA in vitro or in vivo [13]. Another possibility is to co-transfect in vitro transcribed RNA from the expression vector and a helper vector expressing the SFV structural protein genes into mammalian host cells for production of replication deficient SFV particles. The reason for replication deficiency relates to the presence of the packaging signal in the nsP2 region of the expression vector, which prevents the packaging of the RNA from the helper vector [14]. Additionally, to reduce the formation of any replication proficient SFV particles, the second generation pSFV-Helper 2 vector was engineered [15]. Furthermore, for total elimination of RNA recombination and generation of replication proficient SFV particles a two-helper system with the capsid and spike protein genes on separated vectors was applied [16]. Finally, a full-length SFV genome with a second 26S subgenomic promoter and gene of interest introduced either downstream of the nsP genes or the structural genes can be applied, keeping in mind that replication proficient particles will be generated, obviously enhancing biosafety risks. In any case, the options and flexibility are excellent as alphavirus vectors can be used for vaccine development in the form of naked RNA replicons, DNA plasmid vectors and recombinant viral particles as described in more detail in the next sections. 

Due to the positive polarity of the flavivirus RNA genome, similar expression vector systems to SFV have been engineered for Kunjin (KUN) virus-based delivery of RNA replicons, DNA plasmids and recombinant KUN particles [17]. However, foreign genes are introduced between the C20 core protein and the E22 envelope protein for expression as a large polyprotein, which will be processed into individual proteins. Introduction of an FMDV-2A protease sequence in the KUN vector will allow removal of remaining KUN flanking regions from the recombinant product [18]. Virus production from KUN vectors has been facilitated by the engineering of a packaging cell line [19]. In addition to KUN vectors, expression systems for West Nile virus [20,21], yellow fever virus [22,23], dengue virus [24,25] and tick-borne encephalitis [26,27] have been engineered.

In the context of MV vectors, the negative polarity of the ssRNA genome has demanded the application of adequate packaging systems [28] and reverse genetics [29]. The gene of interest can be introduced into the MV expression vector between the phosphoprotein (P) gene and the matrix protein (M) gene or between the HA gene and the large protein (L). Recombinant MV particles are generated by transfection of a helper cell line with plasmids carrying MV constructs and the MV polymerase gene, followed by harvesting of MV particles three days later.

In case of rhabdoviruses, expression vectors have been engineered for both rabies virus (RBAV) [30] and vesicular stomatitis virus (VSV) [31] using reverse genetics based on a recombinant vaccinia virus vector. Engineering of VSV vectors with the N, P and L genes inserted downstream of a T7 RNA polymerase promoter and an internal ribosome entry site (IRES) allowed recovery of DNA-transfected VSV particles in a vaccinia-free system [32]. Similar vectors have been constructed for RABV by insertion of the gene of interest between the RABV N and P genes [33]. A vaccinia-free reverse genetics system has also been engineered for RABV [34].

## 3. Vaccines Against Infectious Diseases

There are numerous examples of preclinical immunization studies against infectious diseases conducted with self-replicating RNA virus vectors, of which some selected examples are presented in Table 1. In the context of Dengue virus (DENV), a single immunization with a tetravalent vaccine based on VEE particles expressing the ectodomain of the DENV E protein (E85) elicited neutralizing antibodies and T cell responses for each of four serotypes tested [35]. Although the immune response was weaker in neonatal BALB/c mice than in adult animals, protective immunity against lethal challenges was obtained after a single vaccine administration. In another approach, the domain III of DENV envelope protein 2 (DV2) was expressed from an MV vector [36]. Immunization of MV-susceptible mice induced robust neutralizing antibodies. In another study on DENV, mice immunized with MV vectors expressing domain II of DV1-4, generated neutralizing antibodies and were protected against four serotypes of DENV [37]. Another flavivirus, ZIKV, was evaluated in a two-vial strategy of highly stable nanostructured lipid carriers (NLCs) and replicon RNA [38]. In this approach, the codon-optimized prM and E genes were introduced into a VEE replicon vector and NLC-RNA complexes, which provided protection against challenges with lethal ZIKV in mice subjected to a single immunization. In another study, a chimeric VSV vector expressing both the membrane-envelope (ME) glycoproteins of ZIKV and Chikungunya (CHIKV) envelope polyprotein (E3-E2-6K-E1) elicited neutralizing antibodies against both ZIKV and CHIKV in wild-type mice and in interferon-receptor-deficient A129 mice [39]. Protection against lethal challenges of both CHIKV and ZIKV was obtained after a single vaccination.

Filoviruses have for obvious reasons received a lot of attention due to such members as Ebola virus (EBOV), presenting urgent needs for the development of novel vaccines especially during the Ebola virus disease (EVD) outbreak in 2014–2016. In this context, KUN virus-like particles (VLPs) expressing the EBOV glycoprotein with the D637L mutation (GP/D637L), which by displaying a cleavage site improved cleavability and shedding of GP, were subcutaneously injected into nonhuman primates [40]. Complete protection was observed in three immunized primates, while one vaccinated and all control animals died. In another approach, it was demonstrated that a recombinant VSV vector expressing the EBOV glycoprotein provided complete protection of macaques challenged with the West African EBOV-Makona strain [41]. A single dose given only seven and three days before challenge resulted in complete and partial protection, respectively. Additionally, VSV-based immunization with EBOV GP provided protection against lethal challenges with three EBOV strains in immunized nonhuman primates [42]. Furthermore, a vaccination study against another filovirus, Marburg virus (MARV), showed protection in nonhuman primates after immunization with VSV-MARV-GP particles [42]. Vaccination against another filovirus, Sudan virus (SUDV), by a single intramuscular administration of VEE particles expressing the SUDV GP provided complete protection in cynomolgus macaques [43]. The immunization did not fully protect the macaques from intramuscular challenges with EBOV. However, intramuscular co-immunization with VEE-SUDV-GP and VEE-EBOV-GP led to complete protection against challenges with both SUDV and EBOV. The intramuscular immunization also resulted in complete protection against challenges with aerosolized SUDV, although two vaccinations were required to reach efficacy. In another study, it was demonstrated that C57BL/6 mice immunized with VEE particles expressing EBOV nucleoprotein (NP) survived challenges with EBOV [44]. Furthermore, immunizations with VEE-EBOV-GP and VEE-EBOV-NP VLPs were evaluated in BALB/c mice and guinea pigs [45]. Vaccination with VEE-EBOV-NP particles showed protection in mice, but not in guinea pigs. In contrast, co-immunization with VEE particles expressing both EBOV-GP and -NP rendered both mice and guinea pigs resistant to EBOV challenges. Recently, immunization with SFV DNA replicons co-expressing EBOV-GP and -VP40, generated both binding and neutralizing antibodies, which were superior to those elicited by a Modified Vaccinia virus Ankara (MVA) vaccine [46]. 

In the context of arenaviruses, development of Lassa virus (LASV) vaccines has been initiated by expressing LASV glycoproteins from VSV vectors [47]. Immunization studies in guinea pigs and macaques provided protection against LASV originating from Liberia, Mali and Nigeria. In another study, the wild-type LASV glycoprotein (GPCwt) and a non-cleavable C-terminally deleted modification (ΔGPfib) expressed from individual VEE 26S subgenomic promoters were immunogenic and protective in immunized mice [48]. The vaccination induced cross-reactive multifunctional T cell responses. Furthermore, a single-cycle replication system for LASV-based replicon particles devoid of the essential LASV GPC gene but supported by a Vero cell line providing the expressed GPC, enabled vaccine propagation [49]. Immunized guinea pigs were protected against challenges with lethal doses of LASV. VEE VLPs have also been applied for expression of Junin virus (JUNV) GPC and Machupo (MACV) GPC eliciting humoral immune responses, which correlated with complete protection against challenges with JUNV and MACV, respectively [50].

Lentiviruses have also been commonly chosen as vaccine targets, obviously because of human immunodeficiency virus (HIV) causing AIDS. For instance, recombinant SFV particles expressing HIV-1 envelope glycoprotein (Env) were applied for immunization studies in mice in comparison to a DNA vaccine and recombinant Env gp160 protein [51]. The highest antibody titers were observed in animals immunized with SFV particles. Furthermore, an SFV RNA replicon vector carrying the HIV-1 Env gene was applied for intramuscular injection into mice [52]. The immunization resulted in induction of Env-specific antibody responses in four out of five mice. Moreover, the monoclonal antibody 12H2 directed against gp41 was produced. In another study, mice were immunized with recombinant SFV particles expressing the Indian HIV-1C env/gag/polRT genes resulting in significant T cell immune responses [53]. The immune responses were stronger for SFV VLPs than RNA replicons. In another approach, BALB/c mice were immunized with plasmid DNA-based SFV vectors expressing Env and a Gag-Pol-Nef fusion protein [54] Initially, immunization studies revealed that lower HIV-specific T cell and IgG responses were obtained for 0.2 μg of SFV DNA compared to a dose of 10 μg, but no differences in immune responses between the doses were detected after boosting with MVA or HIV gp40 protein. It was also confirmed that immunization with the lower dose of SFV DNA elicited superior immune responses in comparison to MVA or HIV gp40 alone. In RNA-based approaches, RNA replicons have been delivered by nanoparticles [55,56]. In this context, the HIV-1 glycoprotein 140 (TV1 gp140) was introduced into the VEE replicon vector containing the 3′ end untranslated region and the packaging signal of SIN and formulated with a cationic nano-emulsion consisting of squalene, 1,2-dioleoyl-3-trimethylammonium-propane (DOTAP) and sorbitan trioleate [55]. The encapsulated RNA replicons intramuscularly administered to rhesus macaques elicited potent cellular immune responses, which were stronger than those observed for VEE VLPs. It was further demonstrated that doses of only 50 µg of encapsulated RNA replicons were safe and provided strong immune responses in primates. In another study, alphavirus RNA replicons expressing the HIV glycoprotein 120 (gp120) were encapsulated in DOTAP-based lipid nanoparticles [56]. Intramuscular administration generated high levels of recombinant protein expression for 30 days in mice compared to only brief and low levels expression of modified conventional mRNA. A single injection resulted in high titers of gp120-specific antibodies. Simian immunodeficiency virus (SIV) has also been subjected to vaccine development. For instance, KUN-based expression of four SIV gag constructs designed for the wild-type SIV gag gene (WT), an RNA-optimized nucleic acid sequence (DX), a human codon-optimized SIV gag gene (OPT) and wild-type matrix and capsid from gag-linked in-frame to reverse transcriptase from pol (Gag-pol) [57]. The Gag-pol vaccine was superior to WT, DX and OPT related to immune response induction and protection of immunized mice against SIV challenges. In another approach, macaques were immunized with SFV and MVA vectors expressing SIV env, gag-pol, nef, rev and tat, which resulted in only low or undetectable cytotoxic T cell responses [58]. In contrast, when macaques were first vaccinated with SFV and boosted with MVA, enhanced antibody responses and high T cell proliferation responses were observed. However, the immunizations did not provide protection against challenges with SIV three months after the last vaccination. In another study in rhesus macaques, immunization was conducted with two injections of VSV particles followed by a single administration of SFV particles expressing SIVsmE660 gag-env alone or in combination with rhesus granular macrophage-colony stimulating factor (GM-CSF) [59]. Two out six animals immunized with SIV gag-pol showed no protection, whereas the combination with GM-CSF resulted in four out of six macaques being infected, suggesting that GM-CSF abrogated protection. Not surprisingly, all animals in the control group receiving influenza virus hemagglutinin protein were infected by SIV. 

Due to the recurrent annual threats of epidemics, influenza viruses have been sought after targets for vaccine development. In this context, intravascular administration of SFV particles expressing the influenza virus nucleoprotein (NP) resulted in systemic immune responses in mice [60]. Furthermore, intranasal delivery elicited mucosal immune responses. In another study, VEE particles expressing the hemagglutinin (HA) gene from the Hong Kong influenza A isolate (A/HK/156/97) provided partial protection against lethal challenges of influenza virus after immunization of chicken of one day of age and complete protection of two weeks old chicken [61]. Additionally, 10 µg of SFV RNA replicons expressing the influenza HA gene induced significant antibody titers after a single intramuscular administration [62]. Moreover, 90% of BALB/c mice immunized twice with SFV-HA RNA replicons were protected against influenza virus challenges, while 90% of control mice died. In comparison to synthetic mRNA, VEE RNA expressing the influenza HA gene required only 1.25 µg self-replicating RNA compared to 80 µg of synthetic mRNA to achieve protection against influenza strains H1N1, H3N2 and B in immunized BALB/c mice [63]. In another approach, the truncated derivative of VEE targeting dendritic cells was used as an adjuvant for inactivated influenza virus (iFlu) [64]. The truncated VEE lacks five nucleotides between the nsP4 stop codon and the beginning of the 3′ end untranslated region (UTR). Immunization of neonatal BALB/c mice elicited strong innate immune responses demonstrating significantly higher and sustained influenza virus-specific IgG antibodies compared to mice immunized with antigen only. Recently, liposome-encapsulation of RNA replicons of the flavivirus classical swine fever virus (CSFV) expressing influenza virus NP elicited immune responses both in vitro and in vivo [65]. 

The current timely coronavirus pandemic has cast a special light on the need of novel vaccines [75,76]. Naturally, vaccine development against SARS-CoV-2, the virus responsible for the COVID-19 pandemic, is currently at an early stage of development although as of May 27, 2020, 115 preclinical studies and 10 clinical trials are in progress according to the WHO [77]. These studies focus on live attenuated viruses, protein subunit vaccines, adenovirus-based delivery and mRNA and mRNA nanoparticles complexes. Two preclinical studies using MV vectors are included. However, previous studies have been conducted for other coronaviruses. In this context, VEE replicon particles were applied for the expression of SARS-CoV S and N proteins from the Urbani strain [66]. Immunization of mice with VEE-SARS-CoV S provided complete short- and long-term protection against challenges with homologues SARS-CoV strains in both young and senescent mice. In contrast, no protection was obtained after immunization with VEE-SARS-CoV N. Related to heterologous SARS-CoV strains, the chimeric icGDO3-S virus encoding a synthetic S gene of the most genetically divergent human GDO3 strain showed strong resistance to neutralization with antisera directed against the Urbani strain. Despite that, immunization with VEE-SARS-CoV S resulted in complete short-term protection against icGDO3-S challenges in young mice, but not in senescent animals. The age-related protection in mice was addressed in a study, where mice were immunized with VEE-SARS-CoV S particles packaged with either attenuated (3014) or wild-type (3000) VEE glycoproteins [67]. It was demonstrated that aged animals immunized with the VEE 3000-based vaccine were protected against SARS-CoV, while immunization with the VEE 3014-based vaccine did not result in survival after the challenge. Furthermore, superior protection was observed in a lethal influenza challenge model. In the context of Middle East respiratory syndrome coronavirus (MERS-CoV), the MERS-S full-length and a soluble variant (MERS-solS) were expressed from a replication-proficient MV vector, which both elicited robust MV- and MERS-CoV-neutralizing antibodies in mice [68]. Vaccination also provided protection against challenges with MERS-CoV.

Among hepatotropic viruses, hepatitis B virus surface antigen (HBsAg) has been expressed from three MV vectors generating humoral responses in MV-susceptible genetically modified mice and rhesus monkeys [69]. However, the differences in HBsAg levels from the MV vectors elicited significantly different HBsAg antibody levels. Moreover, only the MV vector with the highest HBsAg levels showed protection and even so in two out of four animals. In another approach, SFV RNA replicons were applied for the expression of the HBV middle surface envelope glycoprotein (MHB) and the core antigen (HBcAg) [70]. The SFV-G-MHB and SFV-G-HBcAg RNAs were packaged into an VSV G envelope and subjected to immunization of mice. The CD8^+^ T cell responses in mice were greater in magnitude and broader in specificity compared to recombinant protein- and DNA-based vaccines. A single injection with the SFV-G-MHB vaccine protected mice from HBV challenges. In contrast, immunization with SFV-G-HBcAg failed to provide protection. In another study, BALB/c mice were intravenously immunized with 10^7^ and 10^8^ SFV particles expressing the HBV small surface (S) protein, which elicited antibodies reacting with both yeast-derived S antigen and patient-derived S antigen [71]. Immunization with SFV particles expressing pre-S1 resulted in production of pre-S1 and S-specific IgG, and pre-S1 antibodies completely neutralized HBV infectivity.

Although commonly used as delivery vectors, alphavirus replicons have also shown potential as vaccine candidates, particularly for VEE, western equine encephalitis virus (WEE) and eastern equine encephalitis virus (EEE) [72]. In each replicon vector, the furin cleavage site between the E2 and E3 envelope proteins was deleted to prevent cleavage of the p62 precursor and generation of infectious particles, followed by RNA in vitro transcription and generation of VLPs with a two-helper system [78] by co-electroporation of replicon and capsid and glycoprotein (GP) helpers. Immunization in combination (VEE/WEE/EEE) or individually elicited strong neutralizing antibody responses in mice. Moreover, mice were protected against subcutaneous or aerosol challenges with VEE, WEE and EEE for 12 months. Combination immunization with VEE, WEE and EEE generated robust neutralizing antibody responses in macaques, and also provided good protection against aerosol challenges with an epizootic VEE virus and a North American variety of EEE, respectively. In contrast, both the WEE replicon and the VEE-WEE-EEE combination elicited poor neutralizing antibodies and weak protection against WEE challenges. In another approach, the genome of the attenuated VEE V4020 strain was rearranged to include two subgenomic promoters for the expression of the capsid and glycoprotein genes, respectively [79]. The full-length V4020 RNA was expressed from the plasmid pMG4020 from a CMV promoter, which elicited high titers of neutralizing antibodies in BALB/c mice. Challenges with wild-type VEE did not kill the vaccinated mice, while all control mice died. In another study the VEE V4020 vaccine was evaluated in intramuscularly immunized cynomolgus macaques [80]. The vaccination induced high levels of virus-neutralizing antibodies and primates were protected against aerosol challenges with wild-type VEE.

## 4. Vaccines Against Cancer 

Cancer therapy and immunotherapy are areas of great potential for self-amplifying RNA virus vectors. Not surprisingly, most types of cancers have been targeted in preclinical settings (Table 2). In the case of brain tumors, a retargeted MV vector with CD46 and signaling lymphocyte activation molecule (SLAM) ablating mutations in the HA protein combined with a single-chain antibody against the epidermal growth factor receptor (EGFR) showed potent antitumor activity against EGFR- or EGFRvIII-overexpressing primary glioblastoma multiforme (GBM) cell lines [79]. Intratumoral administration in orthotopic GBM12 xenografts resulted in tumor regression and significantly prolonged survival. Alphavirus vectors have also been applied for cancer therapy. In this context, SFV vectors expressing endostatin in comparison to SFV-LacZ and retrovirus-based endostatin expression provided superior inhibition of tumor growth and reduction in intratumoral vascularization in mice with B16 brain tumor xenografts [80]. Moreover, dendritic cells (DCs) transduced with SFV particles expressing interleukin-18 (IL-18) were administered intratumorally into mice bearing B16 brain tumors in combination with systemic administration of IL-12 [81]. The SFV-based treatment resulted in enhanced induction of T helper type 1 responses from tumor specific CD4^+^ and CD8^+^ T cells and natural killers and antitumor immunity. In another study, neuron-targeting micro-RNA sequences (miRT124) introduced into the SFV4 strain generated increased oncolytic potency in human glioblastoma cell lines and resulted in virus replication in tumors, significant inhibition of tumor growth and prolonged survival after intraperitoneal administration in C57BL/6 mice implanted with CT-2A orthotopic gliomas [82]. A chimeric vector was engineered, where the VSV G protein was replaced by the CHIKV envelope proteins (E3-E2-6K-E1) [83]. The VSVΔG-CHIKV vector selectively infected and eliminated tumors and the survival of tumor-bearing mice was extended from 40 to 100 days. Moreover, the chimeric virus also targeted intracranial xenografts from melanoma patients and intravenous administration only infected tumor cells and selectively infected mouse melanoma cells within the brain.

In the context of brain cancer, an MV vector based on the Edmonston strain expressing the carcinoembryonic antigen (CEA) generated cytotoxic effects in breast cancer cell lines and provided significant delay in tumor growth and prolonged survival in BALB/c mice [84]. In another study, the rat HER2/neu gene expressed from SIN and Adenovirus (Ad) vectors inhibited A2L2 tumor cell growth in immunized mice [85]. On the other hand, if immunization took place two days after tumor challenge it was ineffective. However, a prime-boost protocol of SIN-neu immunization followed by Ad-neu administration significantly improved survival rates in mice, which were intravenously challenged with tumor cells. In another study on the A2L2 breast cancer cell line, SIN DNA replicons expressing HER2/neu elicited strong antibody responses and immunized mice showed reduced tumor incidence and reduced tumor mass [86]. Furthermore, intradermal immunization provided tumor protection and required 80% less plasmid DNA compared to conventional DNA plasmids. Additionally, the incidence of lung metastasis was also reduced in vaccinated mice. Related to DCs, VEE particles expressing a truncated neu gene elicited strong neu-specific CD8^+^ T cell and anti-neu IgG responses in mice [87]. Furthermore, a single immunization with DCs transduced with VEE-neuΔ generated regression of large neu tumors in mice. In another study, SFV particles and RNA replicons were administered to tumor-free mice and mice implanted with 4T1 mammary tumors for distribution studies of the luciferase reporter gene [88]. Intravenous RNA replicon administration showed primary brain targeting both in tumor-free and 4T1 tumor bearing animals, while intratumoral injection resulted in high expression levels in tumors. Interestingly, tumor targeting was observed for SFV particles when reduced intravenous or intraperitoneal viral doses were applied. The study also demonstrated that RNA replicons can be successfully re-administered to prolong luciferase reporter gene expression without the risk of immune responses associated with viral delivery. However, neither SFV particle nor RNA replicon delivery enhanced transgene expression from what was observed after the primary administration. In a combination therapy approach, SFV particles expressing IL-12 were administered together with an aroC (-) strain of *Salmonella typhimurium* (LVR01) into 4T1 tumor nodules orthotopically implanted in mice [89]. The treatment resulted in complete inhibition of lethal lung metastasis and long-term survival in 90% of mice after tumors were surgically resected. Although SFV-IL-12 alone showed anti-angiogenic effect, inhibited tumor growth and prolonged survival, the prevention of distant metastasis was related to the synergistic effect of SFV-IL-12 and LVR01. Despite the antitumor potential of LVR01 alone, the combination therapy was superior. Moreover, the order of administration was important as the therapeutic effect was only achieved when SFV-IL-12 was administered first, while pretreatment with LVR01 suppressed the anti-angiogenic effects of SFV-IL-12. In another study on SFV-IL-12, inhibition of tumor growth and lung metastases was demonstrated in a metastatic 4T1 mouse tumor model [90]. 

Cervical cancer has been frequently addressed in search for vaccines based on alphavirus vectors. For instance, immunization of mice with VEE particles expressing the human papilloma virus-16 (HPV-16) E7 protein generated CD8^+^ T cell responses and prevented tumor development [91]. In another approach, immunization of mice with an SFV vector engineered with the translation enhancer signal from the SFV capsid gene and an HPV E6-E7 fusion protein resulted in tumor regression and complete elimination of established tumors [92]. Therapeutic antitumor immunity was established in mice after the combination of intradermal administration of an SFV DNA replicon expressing HPV E6/7 [93] and electroporation. In comparison to conventional DNA plasmid vectors, a 200-fold lower equimolar dose of 0.05 µg resulted in 85% of immunized mice becoming tumor-free. Alphavirus-based immunization has also been combined with local low-dose irradiation, which resulted in 10-fold increase in CD8^+^ T cells in tumors [94]. It was demonstrated that irradiation upregulated chemokines and the combination enhanced antitumor activity. In a triple combination regimen, SFV-HPV E6,7 immunization was combined with administration of 40 mg/Kg sunitinib and low-dose tumor irradiation, which strongly enhanced immunotherapeutic antitumor activity resulting in tumor growth inhibition and 100% tumor-free survival of immunized mice [95].

The classic example of alphavirus-based colon cancer therapy comprises immunization of mice bearing CT26 colon tumors with SFV-LacZ replicon RNA [96]. Antigen-specific antibody and CD8^+^ T cell responses were observed after a single intramuscular injection of 0.1 µg SFV-LacZ RNA. Moreover, pre-immunization provided protection against tumor challenges and therapeutic efficacy and prolonged survival were obtained in mice with pre-existing tumors. In another study, mice implanted with CT29 colon tumors and 4T1 metastasizing breast tumors were immunized with SFV particles expressing the vascular endothelial growth factor receptor-2 (VEGFR-2) [97]. The outcome was inhibition of tumor growth, reduction in tumor angiogenesis and prevention of the spread of metastases. However, co-immunization with SFV-VEGFR-2 and SFV-IL-12 particles resulted in inferior immune responses and reduced inhibition of tumor growth. In contrast, immunization with the combination of SFV-VEGFR-2 and SF-IL-4 particles generated higher anti-VEGFR-2 antibody titers and resulted in prolonged survival. IL-12 expressed from an SFV vector containing the capsid translation enhancement signal showed high efficacy in the CT26 mouse tumor model [90]. It was demonstrated that SFV-based IL-12 expression induced immune cell stimulation and tumor necrosis. In another study, SIN particles expressing LacZ were administered to the mouse colon cancer CT26.CL25 model, showing potent therapeutic effect against existing tumors [98]. In addition to alphavirus vectors, noncytopathic KUN vectors expressing the granulocyte colony-stimulating factor (G-CSF) were subjected to intratumoral administration of mice implanted with CT26 colon tumors and B16-OVA melanomas [99]. The results from the study revealed that tumor regression was associated with the induction of anticancer CD8^+^ T cells and cure was achieved in more than 50% of immunized mice. Moreover, KUN-based immunization resulted in regression of CT26 lung metastases.

In the context of lung cancer, SFV-EGFP particles were demonstrated to induce cell death in human H358a non-small cell lung cancer (NSCLC) cells and also prevented growth of developing H358a spheroids [100]. Moreover, intratumoral administration of SFV-EGFP of nu/nu mice bearing H358a tumor xenografts induced apoptosis resulting in complete tumor regression in three out of seven mice. In another approach, nude mice with implanted A549 adenocarcinoma lung cells were locally injected with replication-competent SFV (VA7)-EGFP particles in comparison to a conditionally replicating Ad vector (Ad5-Delta24TK-GFP) [101]. SFV-based therapy resulted in superior survival rates in mice compared to Ad-based delivery. However, systemic administration was unable to elicit significant immune responses due to low efficacy of intratumoral replication. As previously described above for colon cancer, SIN-LacZ particles also enhanced therapeutic effects in a lung cancer model [98]. In this context, the efficacy was not dependent on tumor cell targeting, but related to the transient β-galactosidase expression in lymph nodes. The immunization produced long-lasting memory T cells and provided protection against challenges with both LacZ-positive and -negative tumor cells. In the case of VSV, NSCLC cell lines were transduced with VSV vectors expressing GFP and interferon-β (IFNβ), respectively, demonstrating oncolytic activity [102]. Moreover, intratumoral administration of VSV-GFP and VSV-IFNβ reduced tumor growth in nude mice with H2009 and A549 xenografts. Intratumoral injection of VSV-IFNβ into syngeneic LM2 lung tumors resulted in tumor regression, prolonged survival and cure of 30% of immunized mice. In the case of MV, the Edmonston strain was evaluated in NSCLC cell lines and immortalized lung epithelial Beas2B cells for viability, induction of apoptosis and viral transgene production by transduction with MV-GFP or MV-CEA [103]. MV transduction resulted in potent killing of most cell lines compared to Beas2B cells. Intratumoral administration of MV-CEA generated tumor regression in nude mice. In another application of MV, the modified rMV-SLAMblind vector unable to bind to its original SLAM receptor showed reduced cell viability in six out of nine human lung cancer cell lines [104]. Moreover, injection into subcutaneous NCI-H441 tumors in severe combined immune deficiency (SCID) mice suppressed tumor growth, and also targeted scattered tumor masses grown in the lungs of xenotransplanted mice.

A number of preclinical studies have been conducted for melanoma. In this context, mice were immunized with a yellow fever virus (YFV) vector expressing the cytotoxic T lymphocyte (CTL) epitope SIINFEKL derived from chicken ovalbumin, which elicited SIINFEKL-specific CD8^+^ lymphocytes and provided protection against challenges with malignant melanoma cells [105]. Furthermore, YFV-based immunization resulted in regression of solid tumors and lung metastases. In the context of alphaviruses, VEE-based expression of tyrosine-related protein-2 (TRP-2) was verified in a B16 mouse melanoma model, which demonstrated humoral immune responses, robust antitumor activity and prolonged survival [106]. In another study, VEE-TRP-2 immunization was combined with administration of immunomodulatory monoclonal antibodies (mAbs) for either antagonist anti-CTL antigen-4 (CTLA-4) or agonist anti-glucocorticoid-induced tumor necrosis factor receptor (GITR) [107]. The VEE-TRP-2 combination with anti-CTLA-4 or anti-GITR mAbs provided complete regression in 50% and 90% of mice, respectively. Studies on melanoma have also been conducted with DNA-based SFV vectors. In this context, VEGFR2 and IL-12 expressed from one SFV DNA vector and survivin and β-hCG antigens expressed from another SFV DNA vector were co-administered and evaluated in a B16 mouse melanoma model [108]. The combination therapy resulted in superior tumor growth inhibition and prolonged survival of mice compared to immunization with either SFV DNA vector alone. Live-attenuated MV strains have been evaluated for their oncolytic activity against a panel of human melanoma cell lines [109]. It was demonstrated that MV resistant melanoma cells possessed a fully functional type I IFN pathway, while sensitive cells showed defects in the pathway. Treatment with ruxolitinib rendered the resistant cells sensitive to MV. It was concluded that type I IFN responses determine whether melanoma cells are sensitive or resistant to MV. Furthermore, the MV Leningrad-16 (L-16) strain showed replication in a panel of human metastatic melanoma cell lines and mediated cell killing in tumor cells [110]. MV L-16 generated statistically significant inhibition of tumor growth in a melanoma xenograft model. Pseudotyping of VSV particles with the non-neurotropic lymphocytic choriomeningitis virus (LCMV) envelope glycoprotein generated VSV-GP vectors capable of efficient killing of mouse, canine and human melanoma cell lines [111]. Immunization of mice with VSV-GP provided prolonged survival in both A375 xenograft and B16-OVA syngeneic mouse models [112]. It has also been demonstrated that priming with intratumoral reovirus followed by boosting with VSV expressing a cDNA library of melanoma antigens (VSV-ASMEL) significantly improved survival of mice with subcutaneous B16 melanoma tumors [113]. When additional boosting with anti-PD1 immune checkpoint blockade was carried out, further enhanced survival and long-term cure was observed.

In the context of ovarian cancer, the above described VSV-GP vector, where the VSV-G protein was replaced by the lymphocytic choriomeningitis virus (LCMV) envelope glycoprotein, showed oncolytic activity both in ovarian cancer cell lines (A2780, HTB77, SKOV6, HOC7, OVCAR3, ID8 and HOSE) and in mice with A2780 tumor xenografts [114]. Combination therapy with ruxolitinib generated superior responses in both subcutaneous and orthotopic xenograft models. In the context of MV, a single-chain antibody (scFv) specific for alpha-folate receptor (FRalpha), overexpressed in 90% of ovarian cancers, was engineered on the MV attachment protein [115]. The tropism and the fusogenic activity of the MV-alphaFR vector was redirected to tumor cells showing no background infectivity of normal human cells. Moreover, tumor volume reduction and overall survival increase were observed in vivo. In the context of alphaviruses, combination therapy of SIN-IL-12 particles and the CPT-11 topoisomerase inhibitor irinotecan resulted in long-term survival in 35% of SCID mice with implanted aggressively growing human ovarian ES2 cancer [116]. In contrast, no long-term survival was observed for monotherapy with either SIN-IL-12 or CPT-11. Moreover, prime-boost immunization approaches with vaccinia virus (VV)-OVA followed by SFV-OVA particle administration or vice versa demonstrated enhanced OVA-specific CD8^+^ T cell immune responses in C57BL/6 mice [117]. In another study, it was demonstrated that SIN-GFP and SIN-Luc vectors induced cellular stress and apoptosis in the MOSEC tumor cell line derived from the ovarian epithelium [118]. In addition, the Maraba virus, belonging to rhabdoviruses, has been used in a prime-boost regimen with the MIS416 microparticle vaccine adjuvant derived from *Propionibacterium acnes* with immunostimulatory muramyl dipeptide and bacterial DNA [119]. Boosting with Maraba virus expressing the human melanoma antigen dopachrome tautomerase (DCT) generated robust tumor-specific CD8^+^ T cell responses resulting in improved tumor control and unique immunological changes in tumors.

Similar to the findings in ovarian tumor cells, SIN vectors expressing GFP and luciferase also induced apoptosis in the pancreatic adenocarcinoma cell line Pan02 [118]. In another approach, VSV vectors expressing GFP were compared to conditionally replicative adenovirus (CRAd), Sendai virus and Respiratory syncytial virus (RSV) for oncolytic activity in 13 pancreatic ductal adenocarcinoma (PDA) cell lines [120]. Generally, VSV demonstrated superior oncolytic ability compared to the other viral vectors tested, although the susceptibility was highly heterogeneous in different PDA cell lines. MIA PaCa2 and Panc-1 cells were highly permissive to both VSV and CRAd, SU86.86 highly permissive only to VSV and HPAF-II showed limited permissiveness to all viruses. A similar pattern was observed in vivo when tested in nude mice. The VSV resistance of HPAF-II cells could be reduced with a combination of polybrene, DEAE-dextran and ruxolitinib, which significantly improved VSV attachment and replication [121]. The triple combination should allow treatment of PDAC tumors highly resistant to oncolytic VSV. In the case of MV vectors, efficient killing of nectin-4 expressing pancreatic cell lines by MV-SLAMblind was observed [122]. Intratumoral injection of MV-SLAMblind showed significant tumor growth suppression in SCID mice implanted with KLM1 and Capan-2 xenografts.

Finally, prostate cancer has also been commonly targeted in preclinical tumor models. For instance, the prostate tumor cell lines PC-3, DU-145 and LnCaP were efficiently killed by MV vectors [123]. Intratumoral administration of MV-CEA vectors generated significant delay of tumor growth and prolonged survival of mice with PC-3 xenografts. In another study, mice and rabbits were immunized with VEE particles expressing the prostate-specific membrane antigen (PSMA) [124]. In comparison to purified PSMA protein, a single dose of 2 × 10^5^ VSV-PSMA particles provided a superior immune response in mice. In another study, mice were immunized with VEE particles expressing the six-transmembrane epithelial antigen of the prostate (STEAP), which elicited specific CD8^+^ T cell responses and resulted in a prolonged survival rate [125]. Moreover, VEE particles expressing the prostate stem cell antigen (PSCA) provided long-term prostate cancer protection in a transgenic adenocarcinoma mouse prostate (TRAMP) model [126]. The survival rate was 90% at 12 months in immunized mice compared to control mice, who had either succumbed to prostate cancer or presented a heavy tumor load. In another study, the prostate-specific antigen (PSA) was expressed from a VEE vector showing efficient infection of mouse DCs in vitro and eliciting a robust PSA-specific response in vivo [127]. Furthermore, immunization with VEE-PSA strongly stimulated production of IgG2a/b anti-PSA antibodies and tumor growth was significantly delayed. The study showed that immunization can overcome immune tolerance to PSA and can mediate rapid clearance of PSA-expressing tumor cells. Related to VSV vectors, the VSV-GP-LCMV expressing luciferase showed great efficacy of infection of prostate cancer cell lines and long-term remission in Du145 and 22Rv1 prostate cancer models after intratumoral injections [127]. Moreover, remission was also observed in syngeneic TRAMP-C subcutaneous tumors and after intravenous vector administration in PC-3M-Luc bone metastases. In a combination therapy approach, oncolytic MV and mumps virus (MuV) vectors were evaluated in a human prostate cancer xenograft model [128]. Greater anti-tumor effect and prolonged survival were obtained from combination therapy compared to individual treatment either with MV or MuV vectors. 

## 5. Clinical Trials

In the context of clinical trials, the number of conducted studies using self-amplifying RNA virus vectors are relatively modest compared to for instance adenovirus and retrovirus vectors (Table 3). In a randomized, double-blind phase I trial, 40 CMV seronegative healthy volunteers were intramuscularly or subcutaneously injected with VEE particles expressing the CMV glycoprotein B or the fusion between CMV pp65 and IE1 proteins [129]. The vaccinations with a lower dose of 1 × 10^7^ infectious units (IU) and a higher dose of 1 × 10^8^ IU were well tolerated with only mild to moderate local reactogenicity, minimal systemic reactogenicity and no clinically important changes in laboratory parameters. Direct IFN-γ responses to CMV antigens were obtained in all vaccinated subjects. The procedure was safe and neutralizing antibodies and multifunctional T cell responses were generated against all three CMV antigens important for protective immunity. In another randomized, double-blind, placebo-controlled phase I trial, HIV-negative volunteers in the US and South Africa were subjected to subcutaneous injection of escalating doses of VEE particles expressing the non-myristoylated form of the HIV-1 subtype C Gag protein [130]. The vaccine was well tolerated with only modest local reactogenicity. However, five serious adverse events were reported, albeit none of them were considered to be related to the vaccine. Disappointingly, the immune responses comprised only low levels of binding antibodies and T cell responses at the highest dose of 1 × 10^8^ IU.

Not surprisingly, EBOV has been the target for numerous vaccine development programs including the application of self-amplifying RNA virus vectors in the light of the EBOV outbreak 2014–2016. For instance, VSV particles expressing the glycoprotein of a Zaire EBOV strain (VSV-ZEBOV) were used for immunization of 78 volunteers with one of three doses (3 × 10^6^, 2 ×10^7^ or 1 × 10^8^ pfu) in a phase I trial to assess safety and immunogenicity [131]. The immunization caused some adverse events such as injection-site pain, fatigue, myalgia and headache, but less adverse events were observed after the second immunization. The antibody titers against ZEBOV glycoprotein were higher for the two highest doses than for the lowest dose. Moreover, the antibody titers were significantly higher after the second vaccination, but the effect diminished after six months. Overall, the immunization elicited anti-EBOV antibody responses and the highest dose should be further evaluated for pre-exposure prophylaxis. A second immunization may further boost the antibody response. In a dose-ranging, observer-blind, placebo-controlled phase I trial, 40 volunteers received doses of 1 × 10^5^, 5 × 10^5^ or 3 × 10^6^ pfu of an attenuated VSV vector, where the VSV G protein was replaced by ZEBOV GP [132]. No serious adverse events occurred, the immunogenicity was comparable across all doses tested and a single immunization elicited sustainable high IgG titers throughout the whole study (180 days). In another phase I trial, 30 healthy volunteers were immunized with 3 × 10^5^, 3 × 10^6^ or 2 × 10^7^ pfu of VSV-ZEBOV [133]. The immunization was well tolerated without any serious vaccine-related adverse events at all doses. EBOV-specific neutralizing antibodies were recorded in nearly all vaccinees and those receiving the highest dose generated EBOV GP-specific T cells. In a follow-up study within the phase I trial, analysis of adaptive immune responses revealed that although no pre-existing immunity was detected, more than one-third of the immunized volunteers developed VSV-specific CTL responses and antibodies [134]. In a dose-finding, placebo-controlled, double-blind phase I/II study, reduction of the VSV-ZEBOV dose from 1–5 × 10^7^ pfu to 3 × 10^5^ pfu improved tolerability, but decreased antibody responses and did not prevent vaccine-induced arthritis, dermatitis or vasculitis [135].

Several phase II/III trials have been conducted for EBOV. For instance, a randomized placebo-controlled phase II study comprising 1500 adults immunized with a chimpanzee Ad3 vector (ChAd3-EBO-Z) and VSV∆G-ZEBOV-GP in Liberia showed adverse events such as injection-site reactions, headache, fever and fatigue [136]. One month after immunization, ChaAd3-EBO-Z and VSV∆G-ZEBOV-GP induced antibody responses in 70.8% and 83.7%, respectively, of vaccinees compared to 2.8% in the control group receiving placebo. After 12 months, 63.5% and 79.5% of the individuals vaccinated with Ad and VSV, respectively, showed immune responses with 6.8% in control subjects. In an open-label, cluster-randomized ring vaccination phase III trial in suspected cases of EVD, 4123 individuals were assigned for immediate vaccination with VSV-ZEBOV and 3528 persons were to receive delayed vaccination [137]. Ten days after randomization, no new cases of EVD were diagnosed in the immediate vaccination group, while 16 EVD cases were registered in the delayed vaccination group confirming a vaccine efficacy of 100%. The interim analysis suggested that the vaccination was safe and efficacious in preventing EVD. In another phase III study in Guinea and Sierra Leone, 2119 individuals were immediately immunized with a single dose of 2 × 10^7^ pfu of VSV-ZEBOV and 2041 subjects were immunized 21 days after randomization [138]. The participants were followed for 84 days showing substantial protection against EVD with no cases of EVD found from day 10 after vaccination. However, the efficacy of vaccination was questioned by claiming of a bias of the medical study team not staying in the communities of the delayed vaccination group, which could be considered as important for disease transmission, even suggesting that EVD was averted due to behavioral changes and the effect of the vaccine was 0% [139]. In response, the team behind the study confirmed that both the immediate and the delayed vaccination communities had similar exposure to health workers during the period after detection and no bias existed [140]. Moreover, at 10 days or more after randomization, eight unvaccinated contacts developed confirmed symptoms of EVD. Therefore, if vaccination had no consequence and the presence of medical staff accounted for the entire effect, the number of cases should be zero or close to zero. In the individually controlled phase II/III trial STRIVE (Sierra Leone Trial to Introduce a Vaccine against Ebola) health care and frontline workers in the five most EVD-affected districts of Sierra Leone received a single intramuscular dose of VSV-ZEBOV at enrollment or 18–24 weeks later [141]. Among the 8673 enrolled participants no EVD cases and no vaccine-related serious adverse events were reported. In a randomized, double-blind, multicenter phase III study in Canada, Spain and the US, individuals were immunized with a single dose or 2 × 10^7^ pfu or 1 × 10^8^ pfu of VSV∆G-ZEBOV-GP and assessed for safety and immunogenicity [142]. The results demonstrated safety of vaccination with no vaccine-related severe adverse events or deaths reported and should support the vaccination of persons at risk of EVD. 

Chikungunya virus, itself a self-amplifying RNA alphavirus, has a history of epidemics in for instance the Republic of Congo [152] and Reunion [153]. A randomized, placebo-controlled, double-blind, phase II clinical trial was conducted in 393 healthy adults in Haiti, Dominican Republic, Martinique, Guadeloupe and Puerto Rico [143]. Two intramuscular injections 28 days apart were well tolerated although 16 serious adverse events unrelated to the vaccine occurred four times in the vaccine group and 12 times in the placebo group. Sixteen mild to moderate adverse events potentially related to the injection procedure occurred 12 times in the vaccine group and four times in the placebo group. Although good safety and tolerability profiles were obtained, clinical efficacy needs to be addressed in phase III trials. In the case of VEE, periodic epidemics have occurred in humans and equines in Latin America making it an important target for vaccine development [154]. A phase I trial was performed by intramuscular and intradermal electroporation of a VEE DNA replicon vector expressing the VEE E3-E2-6K-E1 genes [144]. Participants received 0.5 mg and 2.0 mg of plasmid DNA intramuscularly or 0.08 mg or 0.3 mg DNA intradermally and monitoring continued for 360 days. No serious adverse events related to the vaccine or the device were recorded and both delivery procedures were considered as acceptable although the acute tolerability of the intradermal delivery was judged to be better than the intramuscular procedure. Intramuscular administration elicited detectable VEE-neutralizing antibodies in all vaccinees in the high and low dose groups after two or three immunizations, respectively. Related to the intradermal route, seven out of eight individuals developed VEE-neutralizing antibodies, while five out of eight subjects generated antibody responses after three vaccine injections. Furthermore, a correlation between DNA dose and the magnitude of VEE-neutralizing antibody responses for both intramuscular and intradermal administration routes could be established. 

In the context of cancer, self-amplifying RNA virus vectors have been subjected to several clinical trials (Table 3). A phase I clinical trial was conducted in patients with recurrent glioblastoma multiforme applying doses ranging from 1 × 10^5^ to 2 × 10^7^ TCID_50_ of MV-CEA [145]. So far, a limited number of patients have received 1 × 10^5^ or 1 × 10^6^ TCID_50_ in the resection cavity showing no dose-limiting toxicity. Additional results from the study are expected shortly. An interesting approach has been to utilize liposome-encapsulated SFV particles expressing IL-12 in a phase I trial in kidney carcinoma and melanoma patients [146]. The encapsulation procedure can provide passive tumor targeting, extend the circulation in the blood and protect recognition by host immune defense systems. The encapsulated LipoVIL12 particles were intravenously administered resulting in transient 10-fold increase in IL-12 plasma levels in patients. A good safety profile was established with no toxicity related to liposomes or SFV particles. The study demonstrated that repeated administration of encapsulated SFV particles did not induce any immune response against SFV. Moreover, the maximum tolerated dose (MTD) was determined as 3 × 10^9^ per m^2^, the limitation being the fever response induced by IL-12 expression. 

In the context of lymphomas, the oncolytic MV-Edm-Zagreb vaccine strain (MVEZ) was administered to five patients with stage IIb cutaneous T-cell lymphoma (CTLC) in an open-label dose-escalation phase I trial [147]. Administration of doses of 10^2^ to 10^3^ of TCID_50_ of MVEZ resulted in no dose-limiting toxicity and good tolerability. Complete regression of one CTCL tumor was obtained in one patient after one treatment cycle of four injections. After two treatment cycles, partial regressions were seen in four out of five patients. Related to myeloma, a phase I trial was conducted in patients with recurrent or refractory multiple myeloma with MV vectors expressing the sodium iodide symporter (NIS) in combination with cyclophosphamide or alone [148]. Preliminary findings from two heavily pre-treated patients with a single injection of 10^11^ TCID50 of MV-NIS resulted in complete remission, which lasted for nine months in one patient and complete resolution of bone marrow plasmacytosis in both patients. Furthermore, NIS-mediated imaging illustrated tumor-specific targeting of MV-NIS [155]. Based on the phase I results, a phase II combination therapy of MV-NIS and cyclophosphamide in recurrent or refractory myeloma patients has been initiated. MV vectors expressing CEA have also been subjected to phase I/II clinical trials in patients with recurrent ovarian cancer [149]. Patients with platinum- and paclitaxel-refractory ovarian cancer confined to the peritoneal cavity received intraperitoneally seven doses of MV-CEA, ranging from 10^3^ to 10^9^ TCID_50_, six times monthly. The maximum MV-CEA dose was limited to manufacturing capability of clinical grade oncolytic MV preparation [156]. The immunization was well tolerated with no dose-limiting toxicity and only mild treatment-related adverse events such as non-neutropenic fever and abdominal discomfort. Repeated administration did not induce increase in anti-measles humoral immunity or development of anti-CEA antibodies. The CEA expression was dose-dependent resulting in elevated serum CEA levels in patients receiving 10^9^ TCID_50_. According to the Response Evaluation Criteria in Solid Tumors (RECIST) assessment of antitumor response [157], stable disease was achieved in 14 patients. In this context, nine out of nine patients were from the 10^7^-10^9^ TCID_50_ dose range, while only five out of 12 patients receiving lower dose levels showed stable disease. The median overall survival was 12.15 months, twice as long as expected in this patient population [158]. However, patients receiving high doses of 10^8^–10^9^ TCID_50_ had a median overall survival of 38.4 months.

In the context of pancreatic cancer, VEE-CEA was applied for intramuscular administration of 4 × 10^7^ to 4 × 10^8^ IU in pancreatic cancer patients in a phase I study [150]. Clinically relevant CEA-specific T cell antibody responses were induced after repeated administration of VEE-CEA. Furthermore, CEA-specific antibodies mediated the antibody-dependent toxicity against tumor cells from human colorectal cancer metastases. The study also demonstrated that immunized patients developing CEA-specific T cell responses showed prolonged survival. Finally, VEE-PSMA particles have been applied in a phase I clinical study in patients with castration resistant metastatic prostate cancer (CRPC) [151]. Immunization with either 0.9 × 10^7^ or 3.6 × 10^7^ IU of VEE-PSMA was well tolerated, but only weak PSMA-specific immune responses were obtained. Despite the failure to generate robust immune responses and clinical benefits, the success in eliciting neutralizing antibodies indicated that dose optimization will further enhance the efficacy of the vaccine. 

## 6. Conclusions

In summary, self-amplifying RNA virus vectors have been subjected to a large number of preclinical studies for both infectious diseases (Table 1) and different types of cancers (Table 2). Moreover, self-replicating RNA viral vectors have also been applied for immunizations against bacterial and parasite infections as previously described [159]. Briefly, immunization with SIN DNA-based 85A antigen (Ag85A) expression provided protection of mice challenged with *Mycobacterium tuberculosis* [160] and SFV VLPs expressing the *Plasmodium yoelii* circumsporozoite protein (CS) likewise against malaria infections [161]. In the context of viral infections, such as EBOV, LASV, HIV and influenza virus, robust immune responses have been induced providing in many cases protection against challenges with lethal doses of pathogenic viruses in animal models. Moreover, a clinical phase I trial in healthy volunteers elicited modest antibodies against HIV-1 [130]. Similarly, neutralizing antibodies were elicited in individuals immunized with VEE in a phase I study targeting VEE [144]. Encouraging results have been obtained from phase III trials on EBOV in Africa demonstrating substantial protection against EVD [137,138]. The current COVID-19 pandemic has naturally shed a new light on viral diseases and has accelerated the development of novel vaccines [162]. 

In the case of cancer vaccines, studies in numerous animal models for various cancer indication have demonstrated tumor regression and prolonged survival. As most of the clinical trials conducted so far have comprised phase I studies, the focus has mainly been on safety and tolerability. Despite that, prolonged survival was established in pancreatic cancer patients treated with VEE-CEA [150], stable disease in ovarian cancer patients immunized with MV-CEA [149] and complete remission in one patient with refractory multiple myeloma after treatment with MV-NIS [148]. However, in many of the studies the immune responses were relatively modest, suggesting that optimization of doses, prime-boosting, vector choice and delivery method/route are necessary.

In the context of self-amplifying RNA virus vectors, their main attractive features comprise the immediate efficient amplification of mRNA directly in the cytoplasm and the flexibility of applying delivery vectors based on RNA replicons, recombinant virus particles or VLPs and layered DNA/RNA vectors. The efficacy of self-amplifying RNA virus vectors has been confirmed for RNA replicons by comparison to immunization with synthetic mRNA, where protection of mice against influenza virus challenges was achieved with 64 times less self-amplifying VEE RNA (1.25 µg) compared to 80 µg of synthetic mRNA [63]. Similarly, 100- to 1000-fold lower doses of DNA replicon SIN-HSV-1-gB compared to conventional DNA plasmids were required to obtain antibody responses and protection against lethal challenges with virus in mice [163]. In the context of cervical cancer, where conventional DNA-based immunization failed to prevent tumor outgrowth, already a 200-fold lower equimolar dose of 0.05 µg of SFV-HPV E6/7 DNA provided complete tumor regression in 85% of immunized mice [93].

In attempts to enhance immune responses for development of better vaccines, attention has also been paid to vector engineering and delivery improvements. Oncolytic viruses have been applied for specific infection of tumor cells, leading to their killing without affecting normal tissues [160]. Another approach involves liposome- and polymer-encapsulation strategies with the goals of tumor targeting and protection against degradation and recognition by the host immune system. For instance, liposome encapsulated SFV particles demonstrated tumor targeting and allowed repeated administration in cancer patients [146]. In another approach, polymer-coating of MV expressing the N, P and L proteins (MV-NLP) of the wild-type MV strain resulted in superior oncolytic activity and improved antitumor activity in mice compared to naked MV-NLP [164]. One area of interest has been the targeting of DCs as described for SFV-IL-18 transduced DCs in mice with B16 xenografts [80] and VEE-neu transduced DCs in a mouse tumor model for breast cancer [86]. Furthermore, RNA replicons of CSFV expressing influenza virus NP complexed with liposomes for delivery to DCs, elicited immune responses both in vitro and in vivo [65]. Another approach relates to combination therapy of self-amplifying RNA virus vectors with drugs and irradiation (Table 2), including tumor eradication with SFV-HPV E6,7 particles, sunitinib and low-dose irradiation [95] or enhanced immune responses of VSV and ruxolitinib co-administration [120]. Much attention has also been paid to delivery safety including engineering of vectors providing the highest safety standards. For instance, point mutations introduced into the SFV p62 precursor sequence prevented the cleavage of p62 into E2 and E3 proteins, which resulted in conditionally infectious particles and reduction of production of replication competent SFV particles [15]. Furthermore, split helper systems have been engineered for SFV [16], SIN [165] and VEE [166] by placing the capsid and envelope genes on separate helper vectors, eliminating the production of replication-proficient particles. 

In conclusion, self-amplifying RNA virus vectors provide the flexibility of utilizing RNA replicons, recombinant VLPs or DNA replicon plasmids for immunization studies leading to robust antibody responses and protection against challenges with lethal doses of pathogens or tumor cells. Although so far, clinical responses have been relatively modest, vector development, dosage optimization and delivery improvement will support the production of efficient vaccines in the near future. These needs are even more crucial today with the current COVID-19 pandemic.

## Figures and Tables

**Figure 1 ijms-21-05130-f001:**
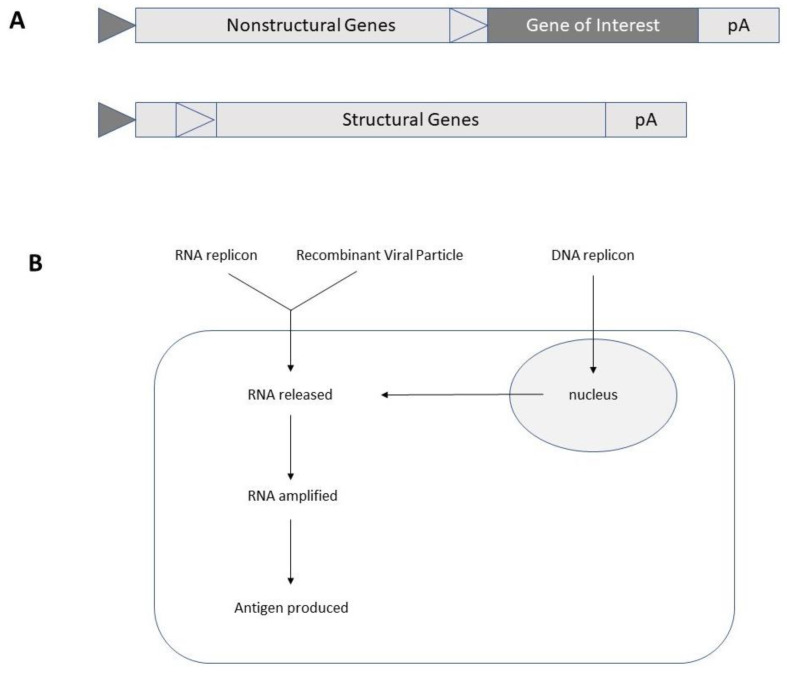
Semliki Forest virus-based expression systems. **A.** Schematic illustration of expression and helper vectors. Dark triangle, SP6 RNA polymerase promoter; light triangle, Semliki Forest virus (SFV) 26S subgenomic promoter. **B.** RNA replicon (naked RNA or liposome-encapsulated RNA), recombinant virus particles or DNA replicon (DNA plasmid) can be used for transfection/infection of cell lines and immunization of animals and humans.

**Table 1 ijms-21-05130-t001:** Examples of preclinical immunizations against viral diseases.

Virus	Target/Antigen	Vector Type	Finding	Ref
**Flaviviruses**				
DENV	E85 ectodomain	VEE VLPs	Dengue protection in mice	[35]
	DV2	MV	Neutralizing antibodies	[36]
	DV1-4	MV	Dengue protection in mice	[37]
Zika virus	prM. E	NLC-VEE RNA	ZIKV protection in mice	[38]
	M, E	VSV VLPs	ZIKV protection in mice	[39]
**Filoviruses**				
EBOV	GP/D637L	KUN VLPs	EBOV protection in 75% of primates	[40]
	EBOV-GP	VSV VLPs	EBOV protection in macaques	[41]
	EBOV-GP	VSV VLPs	EBOV protection in primates	[42]
	EBOV-GP	VSV-VLPs	EBOV protection in macaques	[43]
	EBOV-NP	VEE VLPs	EBOV protection in mice	[44]
	EBOV-GP, NP	VEE VLPs	EBOV protection in mice and guinea pigs	[45]
	EBOV-GP, VP40	SFV DNA	Neutralizing antibodies	[46]
MARV	MARV-GP	VSV VLPs	MARV protection in primates	[42]
SUDV	SUDV-GP	VEE VLPs	SUDV protection in macaques	[43]
**Arenaviruses**				
LASV	LASV-GPC	VSV VLPs	LASV protection in guinea pigs	[47]
	LASV-GPC/ΔGfib	VEE VLPs	LASV protection in mice	[48]
	LASV-GPC	LASV VLPs	LASV protection in guinea pigs	[49]
JUNV	JUNV-GPC	VEE VLPs	JUNV protection in guinea pigs	[50]
MACV	MACV-GPC	VEE VLPs	MACV protection in guinea pigs	[50]
**Lentiviruses**				
HIV-1	HIV-1 Env gp100	SFV VLPs	Humoral immune responses	[51]
	HIV-1 Env	SFV RNA	Antibody responses, mAbs	[52]
	Env/Gag/PolRT	SFV VLPs/RNA	Antigen-specific responses: VLPs > RNA	[53]
	Env/GagPolNef	SFV DNA	Superior to MVA, HIV gp40	[54]
	TV1 gp140	VEE* RNA-NPs	Immunogenicity in macaques	[55]
	Env gp120	VEE RNA-NPs	gp120-specific antibodies	[56]
SIV	Gag-pol	KUN VLPs	SIV protection in mice	[57]
	Env, Gag-pol, Nef, Rev, Tat	SFV + MVA VLPs	Humoral and cellular responses	[58]
	Gag-pol	VSV + SFV VLPs	Partial SIV protection in macaques	[59]
**Influenza**				
Influenza	NP	SFV VLPs	Mucosal immune response	[60]
	HA	VEE-VLPs	Protection in chicken	[61]
	HA	SFV RNA	Protection in mice	[62]
	HA	VEE RNA	Protection in mice	[63]
	iFlu	VEE VLPs	Enhanced immune response	[64]
	NP	CSFV RNA-NPs	Immune response in mice	[65]
**Coronaviruses**				
SARS-CoV	SARS-CoV S	VEE VLPs	Protection in mice	[66]
	SARS-CoV S	VEE VLPs	Protection in mice	[67]
MERS-CoV	MERS-CoV S	MV	Protection in mice	[68]
**Hepatotropic**				
HBV	HBsAg	MV	Partial protection	[69]
	MHB, HBcAg	SFV-G VLPs	Protection in mice by MHB	[70]
	HBV S	SFV VLPs	Neutralization of HBV infectivity	[71]
**Alphaviruses**				
CHIKV	E1-E3, C	VSV	CHIKV protection in mice	[39]
VEE	VEE Replicon	VEE VLPs	Protection in mice, macaques	[72]
EEE	EEE Replicon	EEE VLPs	Protection in mice, macaques	[72]
WEE	WEE Replicon	WEE VLPs	Weak protection in mice, macaques	[72]
VEE	VEE V4020	VEE DNA	VEE protection in mice	[73]
	VEE V4020	VEE DNA	VEE protection in macaques	[74]

CHIKV, Chikungunya virus; CSFV, Classical swine fever virus; DENV, Dengue virus; EBOV, Ebola virus; EEE, Eastern equine encephalitis virus; JUNV, Junin virus; HA, hemagglutinin; HBV, hepatitis B virus; HBcAg, hepatitis B core antigen; HBsAg, hepatitis B surface antigen; HIV, human immunodeficiency virus; iFlu, inactivated influenza virus; LASV, Lassa virus; MACV, Machupo virus; MARV, Marburg virus; mAbs, monoclonal antibodies; MERS-CoV, Middle East respiratory syndrome-coronavirus; MHB, Middle surface HBV glycoprotein; NLC, nanostructured lipid carrier; NP, nucleoprotein; SARS-CoV, severe acute respiratory syndrome-coronavirus; SFV, Semliki Forest virus; SFV-G, Semliki Forest virus with VSV G envelope; SUDV, Sudan virus; VEE, Venezuelan equine encephalitis virus; VEE*, VEE vector containing SIN 3′ untranslated and packaging signal sequences; VLPs, virus-like particles; VSV, vesicular stomatitis virus; WEE, Western equine encephalitis virus; ZIKV, Zika virus.

**Table 2 ijms-21-05130-t002:** Examples of preclinical immunizations targeting cancers.

Cancer	Target/Antigen	Vector	Response	Ref
**Brain**				
GBM	SLAM, EGFR	MV	Tumor regression, prolonged survival	[79]
GBM	Endostatin	SFV VLPs	Tumor regression, prolonged survival	[80]
GBM	IL-18, IL-12	DC-SFV	Enhanced antitumor activity	[81]
GBM	miR124	SFV4	Viral replication in tumors, tumor growth inhibition, prolonged survival	[82]
GBM	Chimeric VLPs	VSVΔG-CHIKV	Tumor cell infection, prolonged survival	[83]
**Breast**				
MDA-MB	CEA	MV	Prolonged survival in mice	[84]
A2L2	HER2/neu	SIN VLPs	Prolonged survival in mice	[85]
A2L2	HER2/neu	SIN DNA	Reduced tumor incidence and mass, protection with 80% less DNA	[86]
A2L2	neuΔ	VEE VLPs + DCs	Specific Abs, tumor regression in mice	[87]
4T1	luciferase	SFV-VLPs, -RNA	Tumor targeting, re-administration	[88]
4T1	IL-12	SFV-IL12, LVR01	Long-term survival, lung metastasis prevention	[89]
4T1	IL-12	SFV VLPs	Tumor regression, metastases	[90]
**Cervical**				
HPV	HPV-16 E7	VEE VLPs	Prevention of tumor development	[91]
HPV	HPV E6-E7	SFV VLPs	Complete elimination of tumors	[92]
HPV	HPV E6/7	SFV DNA	85% of immunized mice tumor-free	[93]
HPV	HPV E6, 7	SFV VLPs, Rad	Enhanced antitumor activity	[94]
HPV	HPV E6, 7	SFV VLPs, Rad+ sunitinib	100% tumor-free survival	[95]
**Colon**				
CT26	LacZ	SFV RNA	Antibody response, protection in mice	[96]
CT26	VEGFR-2	SFV VLPs	Inhibition of tumor growth	[97]
CT26	VEGFR-2 + IL-4	SFV VLPs	Superior survival of mice	[97]
CT26	IL-12	SFV VLPs	Tumor cell necrosis	[90]
CT26	LacZ	SIN VLPs	Therapeutic effect in mouse model	[98]
CT26	G-CSF	KUN VLPs	Cure in >50% of immunized mice	[99]
**Lung**				
H358a	EGFP	SFV VLPs	Complete regression in 3 out of 7 mice	[100]
A549	EGFP	SFV (VA7)	Prolonged survival in mice	[101]
CT26.CL25	LacZ	SIN VLPs	Protection against tumor challenges	[98]
H2009, A549	GFP, IFNβ	VSV VLPs	Reduced tumor growth in mice	[102]
LM2	IFNβ	VSV VLPs	Prolonged survival, cure of 30% of mice	[102]
A549, LLC	GFP, CEA	MV	Tumor regression in mice	[103]
NCI-H441	EGFP	MV-SLAMblind	Suppression of tumor growth in mice	[104]
**Melanoma**				
B16-OVA	G-CSF	KUN VLPs	Cure in >50% of immunized mice	[99]
B16-OVA	SIINFEKL	YVF VLPs	Protection against malignant melanoma	[105]
B16	TRP-2	VEE VLPs	Prolonged survival in mice	[106]
B16	TRP-2	VEE VLPs + anti- CTLA-4/GITR	Tumor regression in mice	[107]
B16	VEGFR-2, IL-12+ Sur, β-hCG	SFV DNA	Prolonged survival in mice	[108]
B16	ASMEL, anti-PD1	Oncolytic VSV + Reovirus	Extended survival, long-term cure in mice	[113]
Mel Z	MV L-16	MV	Inhibition of tumor growth	[110]
A375, B16	GFP, Luc	VSV-GP-LCMV	Prolonged survival in mice	[112]
**Ovarian**				
A2780	Luc	VSV-GP-LCMV	Oncolytic activity in vitro and in vivo	[114]
ARH77	EGFP	MV-alphaFR	Increased survival rate in vivo	[115]
ES2	IL-12	SIN VLPs, CPT-11	Long-term survival in SCID mice	[116]
MOSEC	OVA	SFV VLPs + VV	Enhanced immune responses in mice	[117]
MOSEC	GFP, Luc	SIN VLPs	Induction of cellular stress, apoptosis	[118]
ID9-mp1, ID8	DCT	Oncolytic Maraba	Robust immune response, tumor control	[119]
**Pancreatic**				
Pan02	GFP, Luc	SIN VLPs	Induction of cellular stress, apoptosis	[118]
Panc-1,	GFP	VSV VLPs	Oncolytic activity in vitro and in vivo	[120]
Su.86.86	GFP	VSV VLPs	Oncolytic activity in vitro and in vivo	[120]
KLM1,	SLAM	MV-SLAMblind	Suppression of tumor growth in mice	[122]
Capan-2	SLAM	MV-SLAMblind	Suppression of tumor growth in mice	[122]
**Prostate**				
PC-3				[122]
LnCaP	CEA	MV	Prolonged survival of mice	[123]
TRAMP-C	PSMA	VEE VLPs	Robust immune response in mice	[124]
TRAMP	STEAP	VEE VLPs	Prolonged survival in mice	[125]
TRAMP-PSA	PSCA	VEE VLPs	90% survival rate in mice	[125]
Du145, 22Rv1	PSA	VEE VLPs	Tumor growth delay in mice	[126]
TRAMP, PC-3	Luc	VSV-GP-LCMV	Long-term remission in mice	[127]
	Luc	VSV-GP-LCMV	Remission in subcutaneous tumors and bone metastases	[127]
PC-3	MV, MuV	MV + MuV	Prolonged survival in mice	[128]

ASMEL, cDNA library of melanoma antigens; CEA, carcinoembryonic antigen; CHIKV, Chikungunya virus; CPT-11, topoisomerase inhibitor irinotecan; CTLA-4, cytotoxic T lymphocyte antigen-4; DC, dendritic cell; DCT, dopachrome tautomerase; EGFP, enhanced green fluorescent protein; EGFR, epidermal growth factor receptor; GBM, glioblastoma; GITR, glucocorticoid-induced tumor necrosis factor receptor; HPV, human papilloma virus; IL-4, interleukin-4; IL-12, interleukin-12; IL-18, interleukin-18; LLC, Lewis lung cancer; Luc, luciferase; MDA-MB, MDA-MB-231 tumor; MOSEC, ovarian surface epithelial carcinoma; MuV, mumps virus; MV, measles virus; MV-L16, Leningrad-16 strain; MV-SLAMblind, MV vector unable to bind to SLAM; Pan02, pancreatic tumor cell line; PSCA, prostate stem cell antigen; PSMA, prostate-specific membrane antigen; Rad, radiation; SFV, Semliki Forest virus; SFV (VA7), attenuated replication-proficient SFV; SIN, Sindbis virus; SLAM, signaling lymphocyte activating molecule; TRAMP-C, transgenic adenocarcinoma of the mouse prostate TRP-2, tyrosine related protein-2; VEE, Venezuelan equine encephalitis virus; VEGFR-2, vascular endothelial growth factor receptor-2; VLPs, virus-like particles; VSV, vesicular stomatitis virus; VSVΔG, VSV where the VSV-G protein replaced with CHIKV envelope; VSV-GP-LCMV, VSV where the VSV-G protein replaced by lymphocytic choriomeningitis virus envelope glycoprotein; YFV, yellow fever virus.

**Table 3 ijms-21-05130-t003:** Clinical trials conducted for self-amplifying RNA virus vectors.

Indication	Vector/Antigen	Phase	Response	Ref
**Infections**				
CMV	VEE-gB/p55-IE1	Phase I	CMV-specific Abs	[129]
AIDS	VEE-HIV-gag	Phase I	Modest antibody responses	[130]
EBOV	VSV-ZEBOV	Phase I	Anti-ZEBOV Abs	[131]
	VSV∆G-ZEBOV	Phase I	Sustainable IgG titers for 180 days	[132]
	VSV-ZEBOV	Phase I	EBOV-specific neutralizing Abs	[133]
	VSV-ZEBOV	Phase I/II	Lower dose, improved tolerability	[135]
	VSV∆G-ZEBOV	Phase II	Ab-response in 80% of vaccines	[136]
	VSV-ZEBOV	Phase III	100% protection against EVD	[137]
	VSV-ZEBOV	Phase III	Substantial protection against EVD	[138]
	VSV-ZEBOV	Phase II/III	No EVD, no vaccine-related AEs	[141]
	VSV∆G-ZEBOV	Phase III	Safe, no vaccine-related AEs	[142]
CHIK	CHIK VLPs	Phase II	Safe, well tolerated	[143]
VEE	VEE DNA	Phase I	VEE-specific neutralizing Abs	[144]
**Cancers**				
Glioblastoma	MV-CEA	Phase I	No dose-limiting toxicity	[145]
Kidney	LipoVIL12	Phase I	Safe, tumor targeting	[146]
Lymphoma	MVEZ	Phase I	Tumor regression	[147]
Melanoma	LipoVIL12	Phase I	Safe, tumor targeting	[146]
Myeloma	MV-NIS	Phase I	Complete remission in one patient	[148]
Ovarian	MV-CEA	Phase I/II	Stable disease (high dose treatment)	[149]
Pancreas	VEE-CEA	Phase I	Prolonged survival	[150]
Prostate	VEE-PSMA	Phase I	Neutralizing Abs against PSMA	[151]

Abs, antibodies; AEs, adverse events; CEA, carcinoembryonic antigen; CHIKV, Chikungunya virus; CMV, cytomegalovirus; EBOV, Ebola virus; EVD, Ebola virus disease; HIV, human immunodeficiency virus; LipoVIL12, liposome-encapsulate SFV expressing IL-12; MV, measles virus; MVEZ, MV Edmonston-Zagreb strain; NIS, sodium iodide symporter; PSMA, prostate-specific membrane antigen; SFV, Semliki Forest virus; VEE, Venezuelan equine encephalitis virus; VLPs, virus-like particles; VSV, vesicular stomatitis virus; ZEBOV, glycoprotein of Zaire EBOV.

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
