# Peer review of "Self-Amplifying RNA Viruses as RNA Vaccines"

_ijms, 2020, doi:10.3390/ijms21145130_

Round 1

Reviewer 1 Report

This paper gives a comprehensive introduction and overview to a special class of nucleic acid-based vaccines. It summarizes in a well written manner the design, function, and the current state of pre-clinical and clinical testing of vaccines made of RNA virus genomes of positive and negative polarity. The author also highlights the importance of such vectors in the light of the current COVID-19 pandemic.

The paper is an important contribution to the field at a time when RNA-vaccines gained more visibility due to the need for a rapid vaccine development against SARS-CoV-2.

There are a few minor corrections that should be addressed by the author:

Page 2, line 57. “All self-amplifying RNA viruses initially express their nonstructural genes resulting in the formation of the RNA replication complex (RNA replicon), responsible for extreme RNA replication in infected host cells [7].” – To my understanding the term in brackets should be “spherule”, if this is meant to be synonymous to “replication complex”.

Page 4, line 147. The author should provide a brief explanation about the biological background for chosing the D637L mutation of the EBOV glycoprotein.

Page 7, line 248. “Additionally, 10 mg of SFV RNA replicons expressing the influenza HA gene induced significant antibody titers after a single intramuscular administration [62].” – I assume a typo, and it should be 10 micro grams.

Page 7, line 254. “In another approach, the truncated derivative of VEE targeting dendritic cells was used as an adjuvant for inactivated influenza virus (iFlu) [64].” – could the author briefly explain the truncation that was used in the vector of this reference?

Page 8, line 303/304. The author should mention the rationale of removing the furin cleavage site between E2 and E3.

Page 11, lines 447, 450. Typo in the abbreviation of yellow fever virus (YFV).

There is overall an inconsistency in abbreviating virus names. For many the author writes CSFV, DENV, SFV including the “V” for virus, but VEE, EEE, WEE, SIN… without “V”. I suggest to be consistent.

Author Response

Dear Reviewer,

Thank you for your valuable comments. Please find below my responses to your comments/suggestions.

Page 2, line 57. “All self-amplifying RNA viruses initially express their nonstructural genes resulting in the formation of the RNA replication complex (RNA replicon), responsible for extreme RNA replication in infected host cells [7].” – To my understanding the term in brackets should be “spherule”, if this is meant to be synonymous to “replication complex”.

R: RNA replicon is the correct term

Page 4, line 147. The author should provide a brief explanation about the biological background for chosing the D637L mutation of the EBOV glycoprotein.

R: A brief explanation has been added.

Page 7, line 248. “Additionally, 10 mg of SFV RNA replicons expressing the influenza HA gene induced significant antibody titers after a single intramuscular administration [62].” – I assume a typo, and it should be 10 micro grams.

R: Corrected as suggested.

Page 7, line 254. “In another approach, the truncated derivative of VEE targeting dendritic cells was used as an adjuvant for inactivated influenza virus (iFlu) [64].” – could the author briefly explain the truncation that was used in the vector of this reference?

R: A brief explanation has been added.

Page 8, line 303/304. The author should mention the rationale of removing the furin cleavage site between E2 and E3.

R: Explanation of the removal of the furin cleavage site added.

Page 11, lines 447, 450. Typo in the abbreviation of yellow fever virus (YFV).

R: Typos have been corrected.

Reviewer 2 Report

This is a well written and timely review of a very important field. The review is extensive, ranging from application of these vaccine vectors for infectious disease to cancer. Despite this broad scope I think it was well balanced and tied together very nicely. There were a few additional topics that could have been added in to the review but I would leave this to the author's discretion. For instance the work from Richard Vile's lab where he encodes neoantigens into VSV (Illet et al Gene Therapy 2017 21-30, Pulido et al Nature Biotechnology 2012:337-43).  Also the concept of prime:boost with Maraba virus (McGray et al JITC 2019 7(1) 189, Atherton et al Cancer Immunology Research 2017:10(847-859); Bridle et al Journal of Immunology 2016: 196 (1) 4587-4595). 

Author Response

Dear Reviewer, 

Thank you for your valuable comments/suggestions. Please find my responses below.

There were a few additional topics that could have been added in to the review but I would leave this to the author's discretion. For instance the work from Richard Vile's lab where he encodes neoantigens into VSV (Illet et al Gene Therapy 2017 21-30, Pulido et al Nature Biotechnology 2012:337-43).  Also the concept of prime:boost with Maraba virus (McGray et al JITC 2019 7(1) 189, Atherton et al Cancer Immunology Research 2017:10(847-859); Bridle et al Journal of Immunology 2016: 196 (1) 4587-4595). 

R: Text has been added for the VSV-based expression of the cDNA library of melanoma antigens and the prime-boost regimen with Maraba virus.

Reviewer 3 Report

Manuscript ID: ijms-867411

The paper by Lundstrom, is a well performed review describing the use of self-amplifying RNA viruses as potential RNA vaccines against many viral infections as well as tumor antigens. Various viral and tumor challenge models have been clearly documented and cathagorized in different virus families and origins of tumors. While reading the manuscript I was wondering why the author did not included bacteria and/or parasite infections as well. I don’t know if there are any studies published where RNA vaccines have been used in bacterial and parasitic infection models but this information would certainly increase the importance and impact of the current review.

Nevertheless, this review is acceptable for publishing.

Author Response

Dear Reviewer,

Many thanks for your valuable comments/suggestions. Please find my responses below.

While reading the manuscript I was wondering why the author did not included bacteria and/or parasite infections as well. I don’t know if there are any studies published where RNA vaccines have been used in bacterial and parasitic infection models but this information would certainly increase the importance and impact of the current review.

  1. A brief description (with two examples and a reference to more detailed review) has been added on vaccine development against bacterial and parasite infections in the Conclusion section.